# Endoscopic Ultrasound View of Pneumatosis Cystoides Intestinalis

**DOI:** 10.3390/diagnostics13081424

**Published:** 2023-04-15

**Authors:** Erika Yuki Yvamoto, Spencer Cheng, Guilherme Henrique Peixoto de Oliveira, João Guilherme Ribeiro Jordão Sasso, Mateus Bond Boghossian, Mauricio Kazuyoshi Minata, Igor Braga Ribeiro, Eduardo Guimarães Hourneaux de Moura

**Affiliations:** Serviço de Endoscopia Gastrointestinal do Hospital das Clínicas HCFMUSP, Departamento de Gastroenterologia, Faculdade de Medicina, Universidade de Sao Paulo, Sao Paulo 05403-010, SP, Brazil

**Keywords:** pneumatosis cystoides intestinalis, colonoscopy, endoscopic ultrasound, endoscopy

## Abstract

Pneumatosis cystoid intestinalis (PCI) is a rare condition, with a worldwide incidence of 0.3–1.2%. PCI is classified into primary (idiopathic) and secondary forms, with 15% and 85% of presentations, respectively. This pathology was associated with a wide variety of underlining etiologies to explain the abnormal accumulation of gas within the submucosa (69.9%), subserosa (25.5%), or both layers (4.6%). Many patients endure misdiagnosis, mistreatment, or even inadequate surgical exploration. In this case, a patient presented acute diverticulitis, after treatment, a control colonoscopy was performed that found multiple rounds and elevated lesions. To further study the subepithelial lesion (SEL), a colorectal endoscopic ultrasound (EUS) was performed with an overtube in the same procedure. For safe insertion of the curvilinear array EUS, an overtube with colonoscopy was positioned through the sigmoid as described by Cheng et al. The EUS evaluation evidenced air reverberation in the submucosal layer. The pathological analysis was consistent with PCI’s diagnosis. The diagnosis of PCI is usually made by colonoscopy (51.9%), surgery (40.6%), and radiological findings (10.9%). Although the diagnosis can be made by radiological studies, a colorectal EUS and colonoscopy can be made in the same section without radiation and with high precision. As it is a rare disease, there are not enough studies to define the best approach, although colorectal EUS should be preferred for a reliable diagnosis.

**Figure 1 diagnostics-13-01424-f001:**
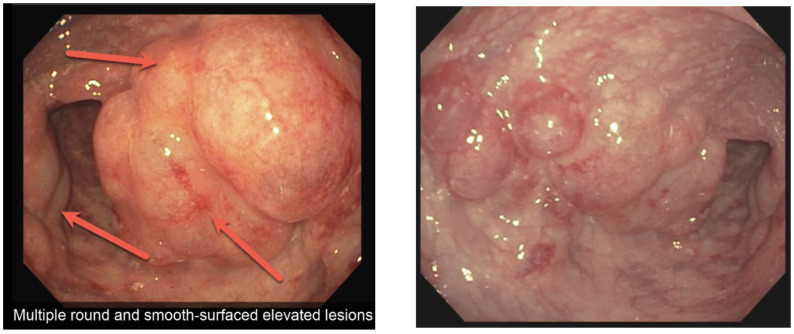
A 45-year-old female patient with a medical history of constipation and recurrent diverticulitis presented herself for an elective colonoscopy eight weeks after her last episode. In the descending colon the red arrows point to multiple rounded and elevated, softened smooth-surfaced, mucosal lesions with foci of enanthem were found without loss of mucosal contiguity. These lesions were homogeneously distributed over the entire circumference of this intestinal segment. PCI is classified into primary (idiopathic) and secondary forms, with 15% and 85% of presentations, respectively [1,2].

**Figure 2 diagnostics-13-01424-f002:**
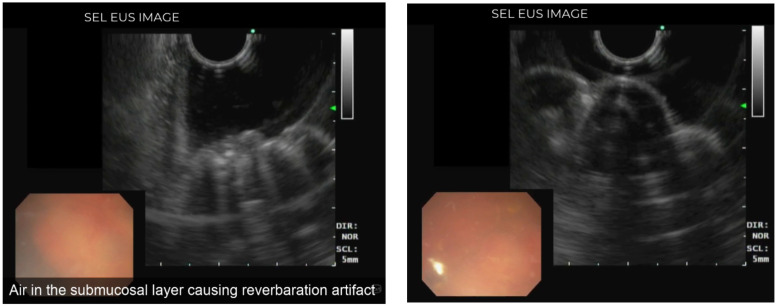
Although the diagnosis can be made by radiological studies [3,4,5,6], a colorectal EUS and colonoscopy can be made in the same section without radiation and with high precision [4,5,6]. An Endoscopic Ultrasound (EUS) (Appendix A) was performed for lesion evaluation. The ultrasound images showed irregular hyperechoic structures, “dirty shadow”, A-lines, and decreased visualization of deeper structures, air reverberation compatible with gas in the submucosal layer, with no nodules or fluid inside.

**Figure 3 diagnostics-13-01424-f003:**
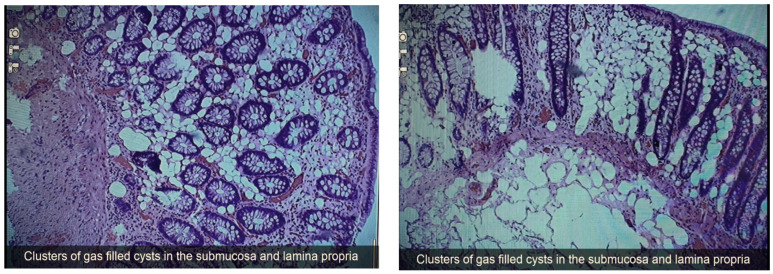
Biopsies of the subepithelial lesions were taken. Pathological analysis, found clusters of gas-filled cysts (as white balls) in the submucosa, and lamina propria were consistent with PCI’s diagnosis. The diverticulitis was the suspected reliable cause.

## Data Availability

No new data were created or analyzed in this study. Data sharing is not applicable to this article.

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
