# Peer review of "Endoscopic Ultrasound View of Pneumatosis Cystoides Intestinalis"

_diagnostics, 2023, doi:10.3390/diagnostics13081424_

Round 1
Reviewer 1 Report
General:
The authors present the interesting case of pneumatosis cystoides intestinalis (PCI) that was diagnosed using endoscopic ultrasonography (EUS). The image of PCI and EUS is good.
However, there are several issues in this paper. Following comments should be referred to improve it.
Comments:
1. There are already several reports on the usefulness of EUS for the diagnosis of PCI, and the novelty is somewhat lacking.
Please refer to the papers as described below.
BMC Gastroenterol. 2019 Nov 6;19(1):176. doi: 10.1186/s12876-019-1087-9.
Endoscopy. 2015;47 Suppl 1 UCTN:E274. doi: 10.1055/s-0034-1391873.
Intern Med. 2001 Sep;40(9):896-900. doi: 10.2169
2. Is the image shown by Figure 2 necessary? Only the image of the overtube is considered to be of little importance.
It can be assumed that the use of an overtube is unnecessary during EUS if a probe type is used.
3. “Introduction1” shows the ”guide to authors”?
Author Response
- There are already several reports on the usefulness of EUS for the diagnosis of PCI, and the novelty is somewhat lacking. Please refer to the papers as described below.
BMC Gastroenterol. 2019 Nov 6;19(1):176. doi: 10.1186/s12876-019-1087-9.
Endoscopy. 2015;47 Suppl 1 UCTN:E274. doi: 10.1055/s-0034-1391873.
Intern Med. 2001 Sep;40(9):896-900. doi: 10.2169
Response: Thank you for your comment.
Following your valuable suggestion, we included the citation of these the papers as [4-6].
“Although the diagnosis can be made by radiological studies[2,4-6], colorectal EUS and colonoscopy can be made in the same section without radiation and with high precision[4-6].”
The reference BMC Gastroenterol. 2019 Nov 6;19(1):176. doi: 10.1186/s12876-019-1087-9 describes one report that performed EUS, but did not provide images or description of technique for introducing EUS and the type of EUS used.
The reference Endoscopy. 2015;47 Suppl 1 UCTN:E274. doi: 10.1055/s-0034-1391873, reported that they used the 12-MHz miniprobe.
And the reference Intern Med. 2001 Sep;40(9):896-900. doi: 10.2169, used the EUS probe catheter.
Different from these previous articles, the current article describes the placement of the overtube for the introduction of another model of EUS, the curvilinear array transducer. In addition, our article presents a video about the case that can illustrate it better. Therefore we consider that our article presents new information comparing to previously published articles and has its importance in clinical management and endoscopic knowledge.
2. Is the image shown by Figure 2 necessary? Only the image of the overtube is considered to be of little importance.
It can be assumed that the use of an overtube is unnecessary during EUS if a probe type is used.
Response: We remove the figure 2 with the overtube image and placed in the text the description of the overtube passage to increase the safety of EUS passage. Different from the miniprobe (Endoscopy. 2015;47 Suppl 1 UCTN:E274. doi: 10.1055/s-0034-1391873.) or catheter probe (Intern Med. 2001 Sep;40(9):896-900. doi: 10.2169), the curvilinear array transducer, due to larger diameter and side vision, needs more caution during passage to avoid laceration and perforation. Thus the use of the overtube becomes safer.
3. “Introduction1” shows the ”guide to authors”?
Response: The ”guide to authors” was placed by mistake in Introduction1. I have removed it from the sentence. Thank you for pointing out the error.
Reviewer 2 Report
Thank you for the opportunity to review this set of interesting images. However, I don't think Figure 2 is necessary at all. The focus of this is the PCI, not the EUS - Figure 2 doesn't add value to the manuscript.
Author Response
Response: Thank you for your comment. We remove the figure 2 with the overtube image and placed in the text the description of the overtube passage to increase the safety of EUS passage.
“To further study the subepithelial lesion (SEL), an colorectal endoscopic ultrasound (EUS) was performed with a overtube in the same procedure. For safe insertion of the curvilinear array EUS, an overtube with colonoscopy was positioned through the sigmoid as described by Cheng et al[4].”
Reviewer 3 Report
The abstract represents the text of the paper.
Introduction of the paper is actually the model.
The author should respect the journal recommendations.
Exemple from the paper:
Introduction:
The introduction should briefly place the study in a broad context and highlight why 27 it is important. It should define the purpose of the work and its significance. The current 28 state of the research field should be carefully reviewed and key publications cited. Please 29 highlight controversial and diverging hypotheses when necessary. Finally, briefly men-30 tion the main aim of the work and highlight the principal conclusions. As far as possible, 31 please keep the introduction comprehensible to scientists outside your particular field of 32 research. References should be numbered in order of appearance and indicated by a nu-33 meral or numerals in square brackets—e.g., [1] or [2,3], or [4–6]. See the end of the docu-34 ment for further details on references.
Author Response
The ”guide to authors” was placed by mistake in Introduction1. I have removed it from the sentence. Thank you for pointing out the error.